# Deboronation-Induced Ratiometric Emission Variations of Terphenyl-Based *Closo*-*o*-Carboranyl Compounds: Applications to Fluoride-Sensing

**DOI:** 10.3390/molecules25102413

**Published:** 2020-05-21

**Authors:** Hyunhee So, Min Sik Mun, Mingi Kim, Jea Ho Kim, Ji Hye Lee, Hyonseok Hwang, Duk Keun An, Kang Mun Lee

**Affiliations:** Department of Chemistry, Institute for Molecular Science and Fusion Technology, Kangwon National University, Chuncheon 24341, Korea; shh6353@naver.com (H.S.); bbcisgj2002@kangwon.ac.kr (M.S.M.); kmg6523@kangwon.ac.kr (M.K.); syoil12@daum.net (J.H.K.); jhlee81@kangwon.ac.kr (J.H.L.); hhwang@kangwon.ac.kr (H.H.)

**Keywords:** *closo***-***o*-carborane, *nido***-***o*-carborane, intramolecular charge transfer, deboronation, color change

## Abstract

*Closo*-*o*-carboranyl compounds bearing the *ortho*-type perfectly distorted or planar terphenyl rings (*closo*-**DT** and *closo*-**PT**, respectively) and their *nido*-derivatives (*nido*-**DT** and *nido*-**PT**, respectively) were synthesized and fully characterized using multinuclear NMR spectroscopy and elemental analysis. Although the emission spectra of both *closo*-compounds exhibited intriguing emission patterns in solution at 298 and 77 K, in the film state, *closo*-**DT** mainly exhibited a π-π* local excitation (LE)-based emission in the high-energy region, whereas *closo*-**PT** produced an intense emission in the low-energy region corresponding to an intramolecular charge transfer (ICT) transition. In particular, the positive solvatochromic effect of *closo*-**PT** and theoretical calculation results at the first excited (S_1_) optimized structure of both *closo*-compounds strongly suggest that these dual-emissive bands at the high- and low-energy can be assigned to each π-π* LE and ICT transition. Interestingly, both the *nido*-compounds, *nido*-**DT** and *nido*-**PT**, exhibited the only LE-based emission in solution at 298 K due to the anionic character of the *nido*-*o*-carborane cages, which cannot cause the ICT transitions. The specific emissive features of *nido*-compounds indicate that the emissive color of *closo*-**PT** in solution at 298 K is completely different from that of *nido*-**PT**. As a result, the deboronation of *closo*-**PT** upon exposure to increasing concentrations of fluoride anion exhibits a dramatic ratiometric color change from orange to deep blue via turn-off of the ICT-based emission. Consequently, the color change response of the luminescence by the alternation of the intrinsic electronic transitions via deboronation as well as the structural feature of terphenyl rings indicates the potential of the developed *closo*-*o*-carboranyl compounds that exhibit the intense ICT-based emission, as naked-eye-detectable chemodosimeters for fluoride ion sensing.

## 1. Introduction

*Closo*-*ortho*-carboranes (1,2-dicarba-*closo*-*o*-dodecaboranes, *o*-1,2-C_2_B_10_H_12_) are well-known boron-cluster components of three-dimensional (3D) icosahedral analogs. Recently, *closo*-*ortho*-carboranes have attracted significant attention as new molecular scaffolds of steric and electronic substituents for luminescent organic and organometallic compounds due to their unique photophysical properties and reasonable thermal and electrochemical stabilities originating from the *o*-carborane unit [1,2,3,4,5,6,7,8,9,10,11,12,13,14,15,16,17,18,19,20,21,22,23,24,25,26,27,28]. These electronic features are imparted by the electron-withdrawing properties of the carbon atoms, and the high polarizability of the σ-aromaticity of the organic and organometallic luminophores that comprise the *o*-carborane moiety. These characteristics lead to the formation of electronic donor-acceptor dyad systems that induce intrinsic intramolecular charge transfer (ICT) transitions between the π-conjugated aromatic groups and the *o*-carborane cage [29,30,31,32,33,34,35,36,37,38,39,40,41,42,43,44,45,46,47,48,49,50,51,52,53]. Such ICT characteristics can induce unique luminescence behavior in various *o*-carborane-based organic luminophores [29,30,31,32,33,34,35,36,37,38,39,40,41,42,43,44,45,46,47,48,49,50,51,52,53,54,55,56,57,58,59,60,61,62,63,64,65,66,67,68,69,70]. Interestingly, such an intramolecular radiative mechanism activated by the ICT transitions in the *o*-carboranyl luminophores has been found amenable to modifications via variations to the structure of the *o*-carborane cages or appended aryl groups [32,43,44,45,46,47,48,49,50,51,52,53,70] and their molecular geometries [66,67,68,69]. Furthermore, the direct control of the ICT-based emission in the *closo*-*o*-carboranyl compounds involves the conversion of *closo*-*o*-carboranes to *nido*-*o*-species (*o*-1,2-C_2_B_9_H_12_^−^, one boron atom removed analog of the *closo*-*o*-carborane cage) by reaction with nucleophilic anions. This powerful process can cause dramatic changes in the inherent electronic environment because of the strong electron-donating property of *nido*-*o*-carboranes [71], leading to the alteration of their luminescent features [72,73,74,75,76,77,78]. For example, Carter et al. reported a fluorene-based dimer bearing an *o*-carborane, which exhibited a visible fluorescence change (orange to bright blue) by degradation to the *nido*-species [73]. Furthermore, Núñez et al. reported photoluminescent *closo*- and *nido*-di-carboranyl and tetra-carboranyl derivatives, which possessed intrinsic ICT electronic transitions and demonstrated differences in the emission band maxima of the two species [75]. We recently reported polyolefins bearing pendant *o*-carborane moieties that exhibit strong blue emissions in the solid state. Notably, the observed emissions disappeared after degradation of the carborane cage upon reaction with hydroxyl ions [76]. Additionally, 1,3,5-tris-(*closo*-*o*-carboranyl-methyl)benzene displayed ratiometric emissive color change via deboronation to the corresponding *nido*-*o*-species [77]. The degradation of the *closo*-*o*-carborane–triarylborane dyad to the *nido*-*o*-carboranyl compound exhibited a turn-on fluorescence response toward fluorides [78]. Thus, these *closo*-*o*-carboranyl derivatives exhibited great potential as polymeric or single-molecular chemodosimeters for sensing nucleophilic anions.

On the basis of inducing significant changes in the electronic properties through the conversion of *closo*-*o*-carborane to its anionic *nido*-*o*-species, we sought to investigate in detail the impact of the deboronation of the *closo*-*o*-carboranyl compounds on their ICT-based emission. For this study, we designed two simple terphenyl-based *o*-carboranyl compounds based on our previous results [69] (Figure 1). The first is 2′,5′-dimethyl-1,1′:4′,1′′-terphenyl-based *closo*-*o*-carborane (**DT**), with distorted *o*-terphenyl rings, which showed a weak ICT-based emission transition, and the second is 6,6,12,12-tetramethyl-6,12-dihydroindeno[1,2-*b*]fluorene-based *closo*-*o*-carboranyl compound (**PT**), with planar *o*-terphenyl rings, which possessed an intense emission from the ICT transition. Subsequently, the designed *closo*-*o*-compounds, as well as the *nido*-*o*-carboranyl compounds, were prepared and fully characterized. The comparison of the photophysical properties of these *closo*- and *nido*-compounds indicated that the deboronation of the *o*-carborane moiety and the structural feature of the appended terphenyl rings may deactivate the ICT transition and also quenching of the emission, thereby providing a novel method for fluoride sensing.

## 2. Results and Discussion

### 2.1. Synthesis and Characterization

The synthetic routes for the terphenyl-based *closo*-*o*- (*closo*-**DT** and *closo*-**PT**) and *nido*-*o*- (*nido*-**DT** and *nido*-**PT**) carboranyl compounds, where the *o*-carborane cages are substituted at both the ends by terphenyl moieties, are outlined in Figure 1. The Sonogashira coupling reaction between ethynyltrimethylsilane and the bromo-precursors **DT1** and **PT1** produced the ethynyltrimethylsilane-substituted terphenyl compounds **DT2** and **PT2**, respectively, in high yields (62% for **DT2** and 83% for **PT2**). The mild base (K_2_CO_3_)-mediated deprotection of the trimethylsilyl protecting groups of **DT2** and **PT2** furnished **DT3** and **PT3**, respectively, which were then subjected to decaborane (B_10_H_14_)-promoted cage-forming reactions in the presence of Et_2_S (Figure 1) [79,80,81] to prepare the *closo*-*o*-carborane-substituted terphenyl compounds *closo*-**DT** and *closo*-**PT**, respectively. The dimethyl groups of *closo*-**PT** were introduced to achieve good solubility in a range of organic solvents. Subsequent treatment of *closo*-**DT** and *closo*-**PT** with excess *n*-tetrabutylammonium fluoride (NBu_4_F, TBAF) in THF at 60 °C led to the conversion of the *closo*-carboranes to the *nido*-species; *nido*-**DT** is the (NBu_4_)_2_-salt of the *nido*-form of *closo*-**DT**, and *nido*-**PT** is the (NBu_4_)_2_-salt of the *nido*-form of *closo*-**PT** (Figure 1).

All of the prepared *closo*- and *nido*-*o*-carboranyl compounds were fully characterized using multinuclear (^1^H{^11^B}, ^13^C, and ^11^B{^1^H}) NMR spectroscopy (Appendix A) and elemental analysis. The ^1^H and ^13^C NMR spectra of *closo*-**DT** and *closo*-**PT** exhibited resonances corresponding to the terphenyl moieties. In addition, five broad singlet peaks were observed between −2 and −15 ppm in the ^11^B{^1^H} NMR spectra of both *closo*-**DT** and *closo*-**PT**, which confirmed the presence of the *closo*-*o*-carborane cage. Furthermore, signals were observed at ~78 and ~61 ppm in the ^13^C-NMR spectra, which were attributed to the two carbon atoms of the *closo*-*o*-carboranyl groups. Unlike the neutral *closo*-**DT** and *closo*-**PT**, the broad singlets (δ = −2.3 and −2.4 ppm) in the ^1^H{^11^B} NMR spectra of both *nido*-**DT** and *nido*-**PT** are characteristic of the B–H–B bridge protons of *nido*-*o*-carborane moieties. The ^11^B{^1^H} NMR signals of *nido*-**DT** and *nido*-**PT** at δ ca. −8 to −37 ppm, which are shifted significantly upfield due to the anionic character of the *nido*-*o*-carboranes, clearly confirmed the presence of the *nido*-*o*-carboranyl boron atoms.

### 2.2. Photophysical Properties of the Closo- and Nido-o-Carboranyl Compounds

The photophysical properties of all terphenyl-based *closo*- and *nido*-*o*-carboranyl were investigated using UV/Vis absorption and photoluminescence (PL) spectroscopies (Figure 2 and Table 1). The *closo*-*o*-carboranyl compounds, *closo*-**DT** and *closo*-**PT**, displayed major absorption bands at λ_abs_ = ~268 and 336 nm, respectively, with structureless vibronic features. These bands were attributed to spin-allowed *π*−*π** LE transitions of the central terphenylene groups [82] and typical ICT transitions between the *o*-carborane units and the central phenyl rings (see the time-dependent density functional theory (TD-DFT) results *vide infra*). Indeed, these ICT-based low-energy absorption bands were not present in the spectra of *nido*-**DT** and *nido*-**PT**, due to which those absorption spectra were slightly blue-shifted (λ_abs_ = 254 and 323 nm, respectively, Table 1) compared with those of the *closo*-compounds. These findings imply that the deboronation of the *o*-carborane cages in the *nido*-species quenches the ICT transitions involving the *o*-carborane unit.

To gain insight into the intrinsic photophysical properties of all *o*-carboranyl compounds, the emissive properties of *closo*-*o*-compounds were examined by PL under a variety of conditions, and further, the emissions of *nido*-*o*-compounds in THF at 298 K were investigated (Figure 2 and Table 1). Although the PL spectra of both *closo*-**DT** and *closo*-**PT** in THF exhibited intriguing emission patterns in all states upon excitation at 292 and 345 nm, respectively, the *closo*-**DT** emission was focused in the high-energy region centered at λ_em_ = ~350 nm, whereas *closo*-**PT** exhibited an intense low-energy emission in the 500 to 600 nm range, which tailed off at 650 nm. With reference to the results of the TD-DFT computational study (vide infra), this high-energy emission appears to originate from the π–π* LE transitions of the central terphenyl moieties. In contrast, the low-energy emission is closely associated with ICT transitions between the *o*-carborane cages and the terphenyl rings. Furthermore, the emission spectrum of *closo*-**DT** in THF at 298 K exhibited an intense emission in the high-energy region at λ_em_ = 350 nm due to *π*−*π** LE transitions based on the central phenyl rings. The fact that the high-energy emission band of *closo*-**DT** was consistently maintained in a variety of solvents of different polarities (λ_em_ = 349–350 nm, Table 1 and Appendix A) and that the low-energy emission of *closo*-**PT** was dramatically altered (Table 1 and Appendix A), strongly indicates that *closo*-**DT** and *closo*-**PT** exhibit LE- and ICT-based emissive characteristics, respectively. These intriguing features are clear evidence that the planarity of the terphenyl rings plays an important role in the alternation of the intramolecular electronic transitions as well as the corresponding radiative decay mechanism [69]. Moreover, *closo*-**DT** exhibited only a trace ICT-based emission in solution (THF solution at 298 K), and the PL spectra in the rigid state (THF at 77 K and in the film state, i.e., 5 wt% doped on poly(methyl methacrylate) (PMMA)) showed an enhanced low energy emission (λ_em_ = 482 nm in THF at 77 K and λ_em_ = 492 nm in the film) that tailed to 550 nm. The emission band for *closo*-**PT** around 500 nm was also significantly increased in the rigid state (THF at 77 K and in film), indicating that the electronic transition for both *closo*-compounds are governed by non-radiative process in solution state at 298 K. This behavior originates from the increased efficiency of the radiative decay associated with the ICT transition in the rigid molecular state, which restricts structural fluctuations such as C–C bond variations in the *o*-carborane cage [9,38,66,67,68,69]. In addition, the PL spectra of the two *nido*-*o*-compounds in THF at 298 K exhibited identical emission patterns in the high-energy region (λ_em_ = 343 nm for *nido*-**DT** and 390 nm for *nido*-**PT**, respectively) alone, and each spectrum of the *closo*-compounds corresponded to the terphenyl-centered *π*−*π** LE transition. Accordingly, these phenomena demonstrate that the CT-based emission can be quenched by the anionic character of *nido*-*o*-carborane as well as the distortion of the terphenyl rings, which inhibits the ICT transitions. Such features suggest that *closo*-*o*-carboranyl compounds that exhibit the intense ICT-based emission, such as *closo*-**PT**, can cause dramatic emission color changes via deboronation of the *o*-carborane cage, owing to the interruption of the ICT transition corresponding to the *o*-carborane and conservation of the LE transition. This phenomenon was verified by spectral changes in the emission of *closo*-**PT** in the presence of TBAF (vide infra).

### 2.3. Computational Chemistry and Orbital Analyses for Closo-o-Carboranyl Compounds

To elucidate the nature of the electronic transitions and to analyze the orbitals of *closo*-**DT** and *closo*-**PT**, their S_0_- and S_1_-optimized structures were subjected to TD-DFT calculations using the B3LYP functional (Figure 3 and Table 2). To include the effects of the THF solvent [83,84], a conductor-like polarizable continuum model was chosen. The computational data for the *S*_0_ state showed that HOMO → LUMO transitions are the major lowest-energy electronic transitions in both *closo*-*o*-carboranyl compounds. The HOMO of each compound is entirely localized on the central terphenyl group (>96%; Appendix A), whereas the orbital contribution of the *o*-carborane unit to each LUMO is slightly higher, at >16%. These results indicate that the lowest-energy absorptions of both *closo*-compounds are attributable to the *π*−*π** transitions on the central terphenyl moieties, with minor contributions from the ICT transitions between the *o*-carborane and terphenyl groups as well. All calculated results based on the optimized *S*_0_ structures are in good agreement with the experimentally observed UV/Vis absorption spectra.

In contrast, the calculated results for the S_1_ states of *closo*-**DT** and *closo*-**PT** indicate that the major transitions associated with the low-energy emissions involve both HOMO → LUMO and HOMO → LUMO+1 transitions (Figure 3 and Table 2). Although the LUMO of each compound is significantly localized on the *o*-carborane moiety (∼80%; Appendix A), each HOMO is predominantly located on the central terphenyl group (>92%). These results strongly suggest that the experimentally observed emissions in the low-energy regions mainly originate from ICT transitions between the *o*-carborane and terphenyl moieties. In addition, each LUMO+1 is mainly located on the central terphenyl group (>86%; Appendix A), strongly indicating that the intense emissions observed in the high-energy region, centered at ~350 nm for *closo*-**DT** and ~370 nm for *closo*-**PT**, originate from *π*−*π** transitions in the terphenyl moieties, i.e., LE-based emissions. Consequently, the electronic transitions that occur in each *o*-carboranyl compound were precisely predicted using computational methods.

### 2.4. Emission-Color Changes of Closo-o-Carboranyl Compounds Via Treatment of Fluoride Anion

Finally, to clarify the changes in the photoluminescence properties exhibited during the conversion of both *closo*-**DT** and *closo*-**PT** to the *nido*-species, we investigated the changes in the emissive patterns of both *closo*-compounds as a function of increasing amounts of TBAF in THF. These conversion processes of both *closo*-compounds to the respective *nido*-species by reaction with the fluoride anion occur consecutively, as clearly evidenced from the changes in the specific peaks of the ^1^H-NMR spectra in THF-*d*_8_ (Figure 4). The aryl protons of both *closo*-compounds in the region from 8.0 to 7.0 ppm shifted steadily to the upfield region upon increasing the concentration of TBAF, and finally, these peaks merged with the corresponding peaks in the spectra of each *nido*-compound in THF-*d*_8_, respectively. In particular, the broad singlet peaks around δ = −2.0 and −2.5 ppm, which were assigned to the B–H–B bridge protons of the *nido*-*o*-carborane, could be gradually monitored by increasing the concentration of TBAF. The results of ^1^H-NMR spectral changes indicate that the conversion of the *closo*-compounds to the *nido*-species almost reached full conversion to that of corresponding pure *nido*-compounds when 5 equivalents of TBAF was used for the deboronation process.

As illustrated in Figure 5, upon addition of incremental amounts of TBAF (0–5 equivalents) into the respective solutions of *closo*-**DT** and *closo*-**PT**, followed by heating at 60 °C for 2 h, the LE-based emission for *closo*-**DT** (λ_em_ = ~350 nm) did not change significantly, whereas the ICT-based emission for *closo*-**PT** (λ_em_ = ~550 nm) underwent gradual quenching, and eventually, a slightly enhanced LE-based emission (λ_em_ ≈ 380–410 nm) remained. In particular, the emission intensities and band shapes of each *closo*-compound after treatment with 5 equivalents of TBAF were mostly similar to those (Figure 5, red-solid lines) of the *nido*-compounds. Consequently, the conversion of *closo*-**PT** to *nido*-**PT** exhibited a vivid emission color change from orange to deep-blue (insets in Figure 5b), whereas *closo*-**DT** did not display any color changes from the emission in spite of the deboronation (insets in Figure 5a). These results demonstrate that degradation to the *nido*-form can not only prevent the ICT transition in the o-carboranyl compounds, but also reinforce the π-π*–LE transition, which induces the emission color changes. Consequently, the luminescence-based color change response due to the alternation of the intrinsic electronic transitions caused by the reaction with fluoride anion and the structural feature of central terphenyl groups, indicates the potential of *closo*-**PT** as a naked-eye-detectable chemodosimeter for fluoride ion sensing.

## 3. Materials and Methods

### 3.1. General Considerations

All operations were performed under an inert nitrogen atmosphere using standard Schlenk and glove-box techniques. Anhydrous solvents (toluene, trimethylamine (NEt_3_), and methanol; Aldrich) were dried by passing through an activated alumina column and stored over activated molecular sieves (5 Å). Spectrophotometric-grade solvents (tetrahydrofuran (THF), toluene, dichloromethane (DCM), methanol, and *n*-hexane) were used as received from Alfa Aesar (Ward Hill, MA, USA). Commercial reagents were used without any further purification after purchase from Sigma-Aldrich (potassium carbonate (K_2_CO_3_), magnesium sulfate (MgSO_4_) St. Louis, MO, USA), bis(triphenylphosphine)palladium(II) dichloride (Pd(PPh_3_)_2_Cl_2_), copper(I) iodide (CuI), diethyl sulfide (Et_2_S), ethynyltrimethylsilane, and poly(methyl methacrylate) (PMMA)). Decaborane (B_10_H_14_) was purchased from Alfa Aesar. The dibromo precursors, 4,4''-dibromo-2',5'-dimethyl-1,1':4′,1''-terphenyl (**DT1**) and 2,8-dibromo-6,6,12,12-tetramethyl-6,12-dihydroindeno[1,2-*b*]fluorene (**PT1**), were prepared as reported in the literature [69]. CD_2_Cl_2_ and THF-*d*_8_, purchased from Cambridge Isotope Laboratories, were dried over activated molecular sieves (5 Å). All nuclear magnetic resonance (NMR) spectra were recorded on a Bruker Avance 400 spectrometer (400.13 MHz for ^1^H, 100.62 MHz for ^13^C, and 128.38 MHz for ^11^B, Bruker, Billerica, MA, USA) at ambient temperature. Chemical shifts are given in ppm and are referenced against external Me_4_Si (^1^H and ^13^C) or BF_3_·Et_2_O (^11^B). Elemental analysis was performed on an EA3000 instrument (Eurovector) at the Central Laboratory of Kangwon National University. UV–Vis absorption and photoluminescence (PL) spectra were recorded on Jasco V-530 (Jasco, Easton, MD, USA) and Horiba FluoroMax-4P spectrophotometers (HORIBA, Edison, NJ, USA), respectively. Fluorescence decay lifetimes (τ_obs_) were measured using a time-correlated single-photon counting spectrometer (FLS920, at the Central Laboratory of Kangwon National University, Edinburgh Instruments Ltd., Livingston, UK) equipped with an EPL 375 ps pulsed semiconductor diode laser as the excitation source and a microchannel plate photomultiplier tube (200–850 nm) as the detector, at 298 K. The absolute PL quantum yields (Ф_em_) were obtained with an absolute PL quantum yield spectrophotometer (HORIBA FluoroMax-4P equipped with an FM-SPHERE 3.2-inch internal integrating sphere, HORIBA, Edison, NJ, USA) at 298 K.

### 3.2. Synthesis of **DT2**

Triethylamine (16 mL) was added via cannulation to a mixture of **DT1** (0.42 g, 1.0 mmol), copper iodide (15 mg), and Pd(PPh_3_)_2_Cl_2_ (62 mg) at 25 °C. After stirring for 15 min, ethynyltrimethylsilane (0.55 mL, 4.0 mmol) was added, and the reaction mixture was heated at 90 °C with stirring for 24 h. After cooling to 25 °C, the volatiles were removed by rotary evaporation to afford a dark brown residue. The crude product was purified by column chromatography on silica gel (eluent: DCM/*n*-hexane = 1/10, *v*/*v*) to yield **DT2** as a yellow solid, 0.28 g (yield = 62%). ^1^H-NMR (CD_2_Cl_2_): δ 7.51 (d, *J* = 8.3 Hz, 4H), 7.32 (d, *J* = 8.4 Hz, 4H), 7.13 (s, 2H), 2.25 (s, 6H, –C*H*_3_), 0.27 (s, 18H, –Si(C*H*_3_)_3_). ^13^C-NMR (CD_2_Cl_2_): δ 142.35, 140.71, 133.04, 132.05, 131.97, 129.60, 121.97, 105.26 (acetylene-*C*), 94.83 (acetylene-*C*), 20.01 (–*C*H_3_), 0.03 (–Si(*C*H_3_)_3_). Anal. Calcd. for C_30_H_34_Si_2_: C, 79.94; H, 7.60. Found: C, 79.87; H, 7.49.

### 3.3. Synthesis of **PT2**

**PT2** was prepared according to a procedure analogous to that used for **DT2**, with **PT1** (0.47 g, 1.0 mmol), copper iodide (15 mg), Pd(PPh_3_)_2_Cl_2_ (62 mg), and ethynyltrimethylsilane (0.55 mL, 4.0 mmol), and was isolated as a yellow solid (0.42 g; yield = 83%). ^1^H-NMR (CD_2_Cl_2_): δ 7.78 (s, 2H), 7.71 (d, *J* = 7.9 Hz, 2H), 7.55 (s, 2H), 7.45 (d, *J* = 7.8 Hz, 2H), 1.53 (s, 12H, –C*H*_3_), 0.27 (s, 18H, –Si(C*H*_3_)_3_). ^13^C-NMR (CD_2_Cl_2_): δ 154.55, 154.16, 140.07, 138.92, 131.40, 126.63, 121.79, 120.08, 114.99, 106.24 (acetylene-*C*), 94.53 (acetylene-*C*), 46.96 (–*C*(CH_3_)_2_), 27.37 (–*C*H_3_), 0.07 (–Si(*C*H_3_)_3_). Anal. Calcd. for C_34_H_38_Si_2_: C, 81.21; H, 7.62. Found: C, 80.99; H, 7.55.

### 3.4. Synthesis of **DT3**

K_2_CO_3_ (0.28 g, 2.0 mmol) was dissolved in methanol (10 mL) and added to a solution of **DT2** (0.23 g, 0.5 mmol) in DCM (5 mL). After stirring for 2 h at 25 °C, the resulting mixture was treated with DCM (50 mL) and the organic layer was separated. The aqueous layer was further extracted with DCM (20 × 2 mL). The combined organic extracts were dried over MgSO_4_, filtered, and evaporated to dryness to afford a white residue. The crude product was purified by washing with *n*-hexane (10 mL) to yield **DT3** as a white solid, 0.13 g (yield = 84%). ^1^H-NMR (CD_2_Cl_2_): δ 7.56 (d, *J* = 8.0 Hz, 4H), 7.34 (d, *J* = 8.0 Hz, 4H), 7.13 (s, 2H), 3.18 (s, 2H, –CC*H*), 2.26 (s, 6H, –C*H*_3_). ^13^C-NMR (CD_2_Cl_2_): δ 142.68, 140.68, 133.06, 132.24, 132.07, 129.66, 120.90, 83.81 (acetylene-*C*), 77.69 (acetylene-*C*), 20.01 (–*C*H_3_). Anal. Calcd. for C_24_H_18_: C, 94.08; H, 5.92. Found: C, 93.77; H, 5.62.

### 3.5. Synthesis of **PT3**

**PT3** was prepared according to a procedure analogous to that used for **DT3** with **PT2** (0.40 g, 0.8 mmol) and K_2_CO_3_ (0.44 g, 3.2 mmol), and was isolated as a white solid (0.25 g; yield = 88%). ^1^H-NMR (CD_2_Cl_2_): δ 7.80 (s, 2H), 7.74 (d, *J* = 7.9 Hz, 2H), 7.59 (s, 2H), 7.50 (d, *J* = 7.8 Hz, 2H), 3.20 (s, 2H, –CC*H*), 1.54 (s, 12H, –C*H*_3_). ^13^C-NMR (CD_2_Cl_2_): δ 154.59, 154.15, 140.34, 138.91, 131.64, 126.88, 120.72, 120.15, 115.06, 84.70 (acetylene-*C*), 77.44 (acetylene-*C*), 46.98 (–*C*(CH_3_)_2_), 27.36 (–*C*H_3_). Anal. Calcd. for C_28_H_22_: C, 93.81; H, 6.19. Found: C, 93.77; H, 6.04.

### 3.6. Synthesis of closo-**DT**

Excess Et_2_S (2.5 equiv., 1.2 mmol) was added at 25 °C to a solution of decaborane (B_10_H_14_, 0.52 mmol) and **DT3** (61 mg, 0.20 mmol) in toluene (20 mL). After heating to reflux, the reaction mixture was further stirred for 72 h. The solvent and volatiles were removed under vacuum and methanol (10 mL) was added. The resulting solid was filtered and redissolved in toluene. The crude product upon washing with *n*-hexane (15 mL), afforded *closo*-**DT** as a white solid (47 mg. Yield = 43%). ^1^H{^11^B} NMR (THF-*d*_8_): δ 7.66 (d, *J* = 8.2 Hz, 4H), 7.38 (d, *J* = 8.1 Hz, 4H), 7.12 (s, 2H), 5.13 (s, 2H, CB-C*H*), 2.54 (br s, 8H, CB-B*H*), 2.39 (br s, 3H, CB-B*H*), 2.30 (br s, 9H, CB-B*H*), 2.23 (s, 6H, –C*H*_3_). ^13^C-NMR (THF-*d*_8_): δ 144.05, 140.59, 133.27, 133.24, 132.34, 130.14, 128.04, 77.58 (CB-*C*), 61.49 (CB-*C*), 19.74 (–*C*H_3_). ^11^B{^1^H} NMR (THF-*d*_8_): δ −4.44 (3B), −6.57 (1B), −10.84 (5B), −12.77 (7B), −14.67 (4B). Anal. Calcd. for C_28_H_38_B_20_: C, 56.92; H, 6.48. Found: C, 56.79; H, 6.33.

### 3.7. Synthesis of closo-**PT**

*Closo*-**PT** was prepared according to a procedure analogous to that used for *closo*-**DT,** with decaborane (B_10_H_14_, 0.52 mmol), **PT3** (78 mg, 0.20 mmol), and Et_2_S (2.5 equiv.). The crude product upon washing with *n*-hexane (15 mL), afforded *closo*-**PT** as a white solid (42 mg, Yield = 35%). ^1^H{^11^B} NMR (THF-*d*_8_): δ 7.94 (s, 2H), 7.81 (d, *J* = 8.1 Hz, 2H), 7.67 (s, 2H), 7.58 (d, *J* = 7.9 Hz, 2H), 5.14 (s, 2H, CB-C*H*), 2.56 (br s, 7H, CB-B*H*), 2.50 (br s, 1H, CB-B*H*), 2.39 (br s, 2H, CB-B*H*), 2.30 (br s, 10H, CB-B*H*), 1.54 (s, 12H, –C*H*_3_). ^13^C-NMR (THF-*d*_8_): δ 155.41, 154.80, 141.67, 138.98, 133.36, 127.39, 122.52, 120.66, 115.65, 78.45 (CB-*C*), 61.55 (CB-*C*), 47.50 (–*C*(CH_3_)_2_), 27.08 (–*C*H_3_). ^11^B{^1^H} NMR (THF-*d*_8_): δ −2.73 (3B), −4.64 (1B), −9.09 (5B), −10.77 (7B), −12.86 (4B). Anal. Calcd. for C_32_H_42_B_20_: C, 59.79; H, 6.59. Found: C, 59.87; H, 6.45.

### 3.8. Synthesis of nido-**DT**

*Closo*-**DT** (0.027 g, 0.05 mmol) was dissolved in 0.3 mL of a 0.2 M solution of *n*-tetrabutylammonium fluoride (TBAF) in THF at 25 °C. The reaction mixture was heated to reflux (60 °C) and stirred for 2 h. After cooling to 25 °C, the resulting mixture was treated with 50mL of distilled water and 50 mL of DCM, and the organic portion was separated. The aqueous layer was further extracted with DCM (20 mL). The combined organic portions were dried over MgSO_4_, filtered, and concentrated to dryness, affording a pale yellow residue. The crude product upon washing with methanol (15 mL), afforded *nido***-DT** as a white solid (26 mg, Yield = 52%). ^1^H{^11^B} NMR: δ 7.28 (d, *J* = 8.0 Hz, 4H), 7.13 (d, *J* = 7.9 Hz, 4H), 7.07 (s, 2H), 3.11 (m, 16H, *n*-butyl-C*H*_2_), 2.36 (s, 2H, CB-C*H*), 2.25 (s, 6H, –C*H*_3_), 2.12 (br s, 4H, CB-B*H*), 1.88 (br s, 4H, CB-B*H*), 1.82 (br s, 4H, CB-B*H*), 1.62 (m, 16H, *n*-butyl-C*H*_2_), 1.43 (m, 16H, *n*-butyl-C*H*_2_), 1.26 (br s, 6H, CB-B*H*), 1.02 (t, *J* = 7.2 Hz, 24H, *n*-butyl-C*H*_3_), −2.36 (br s, 2H, B-*H*-B). ^13^C NMR (CD_2_Cl_2_): δ 144.65, 140.78, 138.40, 132.83, 132.21, 128.64, 126.71, 59.43 (*n*-butyl-*C*H_2_), 24.29 (*n*-butyl-*C*H_2_), 20.18 (–*C*H_3_), 20.10 (*n*-butyl-*C*H_2_), 13.76 (*n*-butyl-*C*H_3_). ^11^B{^1^H} NMR (CD_2_Cl_2_): δ −8.97 (3B), −10.43 (2B), −13.79 (1B), −18.28 (3B), −19.50 (1B), −23.00 (1B), −32.95 (3B), −36.10 (4B). Anal. Calcd. for C_60_H_110_B_18_N_2_: C, 68.37; H, 10.52; N, 2.66. Found: C, 68.11; H, 10.42; N, 2.54.

### 3.9. Synthesis of nido-**PT**

A procedure analogous to that for *nido***-DT** was employed using *closo*-**PT** (0.027 g, 0.04 mmol) and 0.23 mL of a 0.2 M solution of TBAF in THF. The crude product, upon washing with methanol (15 mL), afforded *nido***-PT** as a white solid (26 mg, Yield = 60%). ^1^H{^11^B} NMR (CD_2_Cl_2_): δ 7.65 (s, 2H), 7.49 (d, *J* = 7.9 Hz, 2H), 7.32 (s, 2H), 7.23 (d, *J* = 7.9 Hz, 2H), 3.08 (m, 16H, *n*-butyl-C*H*_2_), 2.39 (s, 2H, CB-C*H*), 2.12 (br s, 4H, CB-B*H*), 2.00 (br s, 1H, CB-B*H*), 1.89 (br s, 5H, CB-B*H*), 1.60 (m, 16H, *n*-butyl-C*H*_2_), 1.48 (s, 12H, –C*H*_3_), 1.41 (m, 16H, *n*-butyl-C*H*_2_), 1.31 (br s, 4H, CB-B*H*), 1.26 (br s, 4H, CB-B*H*), 1.00 (t, *J* = 7.2 Hz, 24H, *n*-butyl-C*H*_3_), −2.34 (br s, 2H, B-*H*-B). ^13^C-NMR (CD_2_Cl_2_): δ 153.80, 153.54, 145.25, 138.62, 136.55, 126.10, 121.40, 118.72, 114.00, 59.40 (*n*-butyl-*C*H_2_), 46.71 (–*C*(CH_3_)_2_), 27.74 (–*C*H_3_), 24.27 (*n*-butyl-*C*H_2_), 20.09 (*n*-butyl-*C*H_2_), 13.75 (*n*-butyl-*C*H_3_). ^11^B{^1^H} NMR (CD_2_Cl_2_): δ −8.91 (3B), −10.45 (2B), −13.68 (1B), −18.46 (3B), −19.47 (1B), −23.09 (1B), −32.95 (3B), −36.05 (4B). Anal. Calcd. for C_64_H_114_B_18_N_2_: C, 69.49; H, 10.39; N, 2.53. Found: C, 69.30; H, 10.16; N, 2.39.

### 3.10. UV/Vis Absorption and Photoluminescence (PL) Experiments

The solution-phase UV–Vis absorption and PL measurements of the *closo*- and *nido*-*o*-carbornyl compounds were performed in degassed organic solvents with a 1 cm quartz cuvette (3.0 × 10^−5^ M) at 298 K. PL measurements for the *closo*-compounds were also performed in THF at 77 K and in the film state (5 wt% doped in PMMA) on 1.5 × 1.5 cm quartz plates (thickness = 1 mm) at 298 K.

### 3.11. Computational Studies

The optimized geometries for the ground (S_0_) and first excited (S_1_) states of both *closo*-*o*-carboranyl compounds (*closo*-**DT** and *closo*-**PT**) in THF were obtained using the B3LYP/6-31G(d,p) [85] level of theory. The vertical excitation energies at the optimized S_0_ geometries as well as the optimized geometries of the S_1_ states were calculated using time-dependent density functional theory (TD-DFT) [86] at the same level of theory. Solvent effects were included using the conductor-like polarizable continuum model (CPCM) [83,84]. All geometry optimizations were performed using the Gaussian 16 program [87]. The percent contribution of a group in a molecule to each molecular orbital was calculated with the GaussSum 3.0 program [88]. Visualizations were prepared using GaussView 6 [89].

## 4. Conclusions

We herein reported the preparation and characterization of distorted and planar terphenyl-based *closo*- (*closo*-**DT** and *closo*-**PT**) and *nido*- (*nido*-**DT** and *nido*-**PT**) *o*-carboranyl compounds. Although *closo*-**DT** exhibited strong π–π* LE-based emission in THF at 298 K in the high-energy region, *closo*-**PT** demonstrated intense emission in the low-energy region that was attributable to the ICT transitions involving the *o*-carborane cage. Interestingly, both *nido*-compounds exhibited LE-based emission alone in the same condition due to the anionic character of the *nido*-*o*-carborane cages, which cannot cause the ICT transitions. Consequently, the successful deboronation of *closo*-**PT** to *nido*-**PT** upon exposure to increasing concentration of fluoride anion leads to ratiometric emission color change from orange to deep-blue in solution. Such results strongly imply that the fine-tuning of electronic and structural features, which can control the ICT-based emission, shows the potential of *closo*-*o*-carboranyl compounds as candidates for naked-eye-detectable chemodosimeters for fluoride ion-sensing.

## Figures and Tables

**Figure 1 molecules-25-02413-f001:**
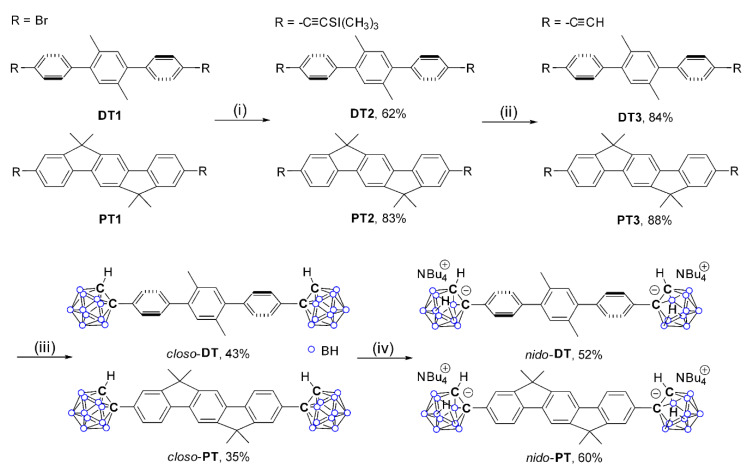
Synthetic routes to the terphenyl-based *closo*- and *nido*-*o*-carboranyl complexes, *closo*-**DT**, *closo*-**PT**, *nido*-**DT**, and *nido*-**PT**. Reaction conditions: (**i**) Ethynyltrimethylsilane, CuI, Pd(PPh_3_)_2_Cl_2_, NEt_3_/toluene, r.t., 24 h. (**ii**) K_2_CO_3_, methanol, r.t., 2 h. (**iii**) B_10_H_14_, Et_2_S, toluene, 110 °C, 72 h. (**iv**) *n*-tetrabutylammonium fluoride (TBAF), THF, 60 °C, 2 h.

**Figure 2 molecules-25-02413-f002:**
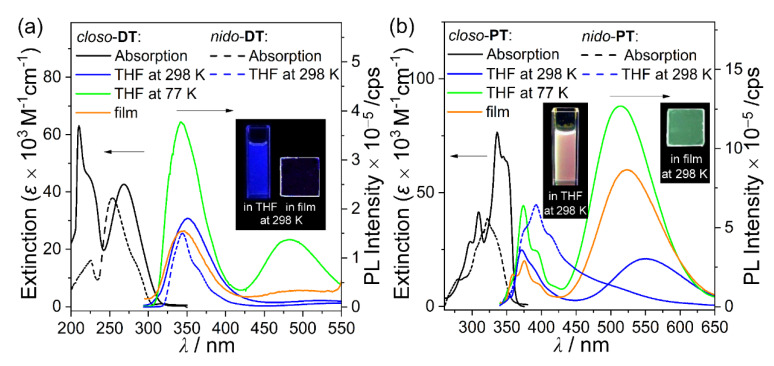
UV–Vis absorption and photoluminescence (PL) spectra for (**a**) *closo*- and *nido*-**DT** (λ_ex_ = 292 nm) and (**b**) *closo*- and *nido*-**PT** (λ_ex_ = 345 nm). Black-solid: absorption spectra in THF (30 μM) for *closo*-species. Black-dash: absorption spectra in THF (30 μM) for *nido*-species. Blue-solid: PL spectra in THF (30 μM) at 298 K for *closo*-species. Blue-dash: PL spectra in THF (30 μM) at 298 K for *nido*-species. Green-solid: PL spectra in THF (30 μM) at 77 K for *closo*-species. Orange-solid: PL spectra of the films (5 wt% doped on PMMA) at 298 K for *closo*-species. Inset figures show the emission color in each state of *closo*-species under irradiation by a hand-held UV lamp (λ_ex_ = 295 nm for *closo*-**DT** and 365 nm for *closo*-**PT**).

**Figure 3 molecules-25-02413-f003:**
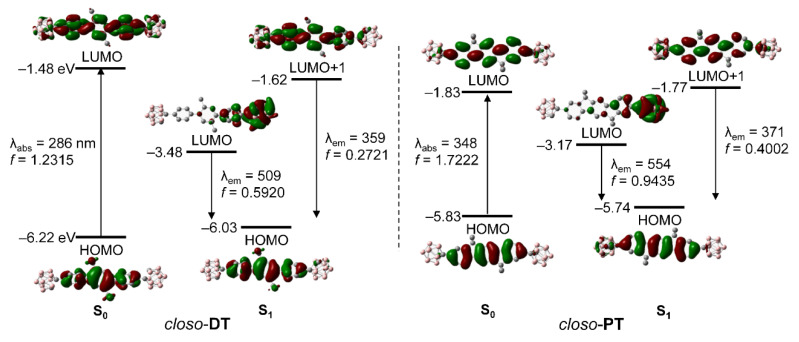
Frontier molecular orbitals of *closo*-**DT** and *closo*-**PT** in their ground states (S_0_) and first excited singlet states (S_1_), and their relative energies calculated by DFT (isovalue = 0.04). The transition energy (in nm) was calculated using the TD-B3LYP/6-31G(d) level of theory.

**Figure 4 molecules-25-02413-f004:**
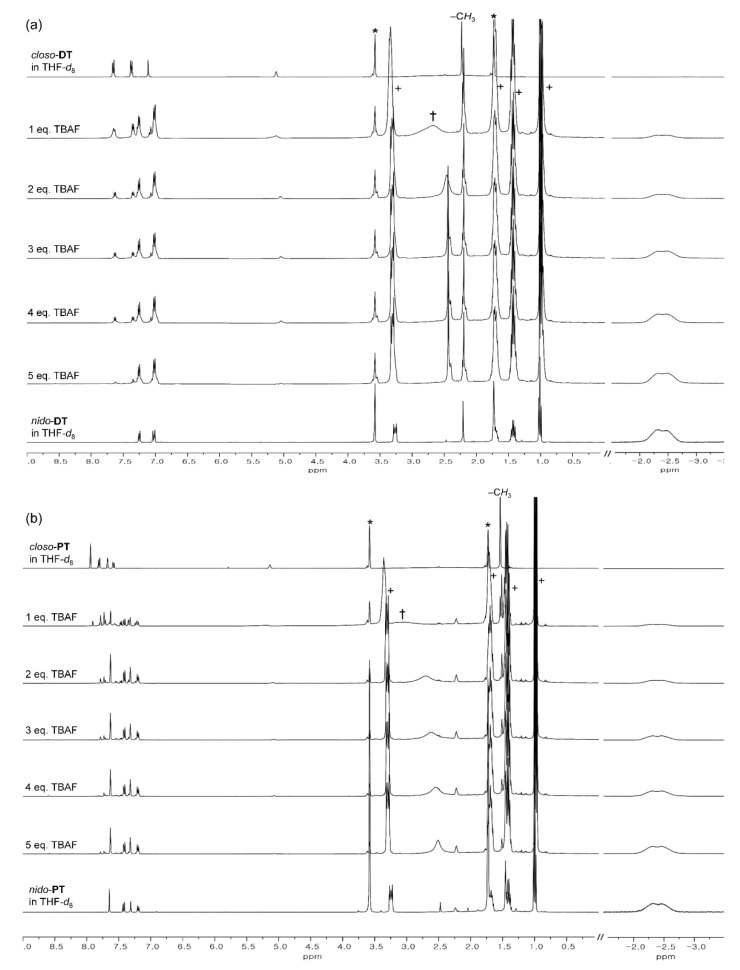
^1^H-NMR spectral changes of (**a**) *closo*-**DT** and (**b**) *closo*-**PT** upon increasing the amount of added fluoride anions and comparison with those of *nido*-**DT** and *nido*-**PT** (∗ from residual THF in THF-*d*^8^, † from *n*-butyl group of excess TBAF, and + from *n*-butyl group for each *nido*-compound).

**Figure 5 molecules-25-02413-f005:**
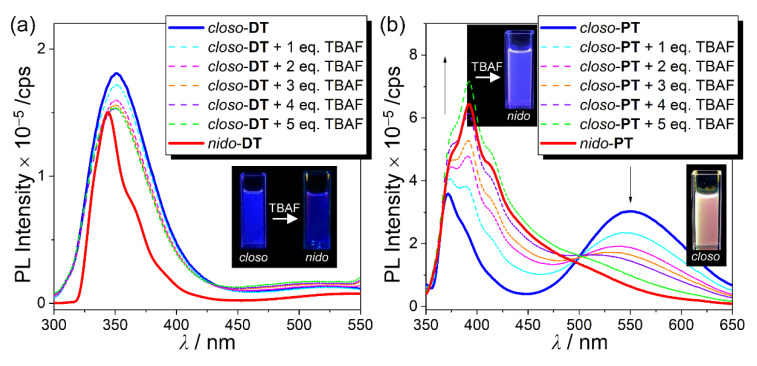
Spectral changes in the emission of (**a**) *closo*-**DT** (3.0 × 10^−5^ M, λ_ex_ = 292 nm) and (**b**) *closo*-**PT** (3.0 × 10^−5^ M, λ_ex_ = 345 nm) in THF in the presence of different amounts of TBAF, upon heating at 60 °C for 2 h. Insets are photographs of each *closo*- and *nido*-type (3.0 × 10^−5^ M in THF) under a UV lamp (λ_ex_ = 295 nm for **DT** derivatives and 365 nm for **PT** derivatives).

**Table 1 molecules-25-02413-t001:** Absorption and emission data for terphenyl-based *o*-carboranyl compounds.

Compound	*λ*_abs_^1^/nm (ε × 10^−3^ M^−1^ cm^−1^)	*λ*_ex_/nm	*λ*_em_/nm				
			Tol ^2^	THF ^2^	DCM ^2^	77 K ^1^	Film ^3^
*closo*-**DT**	268 (42.6)	292	349	350	349	343, 482	345, 492(sh)
*nido*-**DT**	254 (37.8)	292	-	343	-	-	-
*closo*-**PT**	336 (84.8)	345	376, 521	374, 549	375, 560	374, 514	375, 524
*nido*-**PT**	323 (38.2)	345	-	390	-	-	-

^1^*c* = 30 μM in THF. ^2^
*c* = 30 μM, observed at 298 K. ^3^ Measured in the film state (5 wt% doped on PMMA) at 298 K.

**Table 2 molecules-25-02413-t002:** Major low-energy electronic transitions in *closo*-**DT** and *closo*-**PT** involving their ground states (S_0_) and first excited singlet states (S_1_) calculated using the TD-B3LYP/6-31G(d) level of theory ^1.^

	State	*λ*_calc_/nm	*f* _calc_	Assignment
*closo*-**DT**	S_0_	285.7	1.2315	HOMO → LUMO (98.0%)
	S_1_	509.37	0.592	HOMO → LUMO (99.6%)
		359.14	0.2721	HOMO → LUMO+1 (87.7%)
*closo*-**PT**	S_0_	348.17	1.7222	HOMO → LUMO (98.8%)
	S_1_	554.44	0.9435	HOMO → LUMO (99.7%)
		371.34	0.4002	HOMO → LUMO+1 (78.9%)

^1^ Singlet energies for vertical transitions were calculated using optimized *S*_1_ geometries.

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
