# Peer review of "Deboronation-Induced Ratiometric Emission Variations of Terphenyl-Based Closo-o-Carboranyl Compounds: Applications to Fluoride-Sensing"

_molecules, 2020, doi:10.3390/molecules25102413_

Round 1

Reviewer 1 Report

The paper by K.M. Lee and co-authors (Reference: molecules-799158) describes the synthesis, characterization, photophysical properties and theoretical studies of four carboranyl-containing terphenyl-based compounds. They use two different terphenyl rings, one perfectly distorted to get compound closo-DT, and a planar o-terphenyl ring to get closo-PT. They analyse their emission properties in solution (solvatochromic effect and different temperatures) and film state, as well as the changes produced in the emission colour after deboronation of the closo cluster to lead to the nido-species. Similar studies have been previously addressed by the group for other carborane-based compounds. The authors have designed these compounds following a previous paper published by the group (Chem. Commun, 2019, ref 63). The synthetic procedure and the characterization of the compounds are properly done. The TD-DFT calculations match well with the experimental data.

Some minor mistakes:

-Page 5, line 226, the nido-DT is not an adduct but a salt of tetrabuthylammonium, since the nido clusters has a negative charge. Consequently, the term adduct should be removed over the manuscript.

- The term “decoupled” is not necessary, in 1H{11B decoupled} or 11B{1H decoupled}, you might just use 1H{11B} and 11B{1H}.

- Why NMR spectra of closo compounds are carried out in THF-d8, and nido in CD2Cl2? Is there any problem of solubility? In fact, the 11B NMR spectra of the nido compouds seem very diluted. On the other hand, the photophysical measurements are performed in THF at 298K. I wonder if authors have tried to get the NMR spectra for nido derivatives in THF-d8.

-Some doubts and question on the photophysical properties:

  • Why the excitation spectrum is so different to the absorption one for closo-DT?, The excitation maximum is 292 nm, whereas there are two maxima in the UV-vis spectrum (220 and 268 nm). To which transitions can be attributed the high energy band at around 220 nm that in fact is the maximum of absorption in the UV-vis of closo-DT? In the case of closo-PT, both absorption and excitation spectra match well than for closo-DT, what is the explanation?
  •  
  • In the table 1 there are not the fluorescence quantum yield values. Do you have determined them? In general, it is well kown that closo-o-carborane causes the quenching of the fluorescence in solution, for that reason it should be interesting to have these values for the four compounds and compare them.
  • Although the authors affirm that both closo derivatives (closo-DT and closo-PT) exhibit dual-emission patterns, it is very difficult for this referee (and I guess for any reader) to observe the low-energy emissions attributable to the ICT transitions for compound closo-DT in Figure 3, whereas for closo-PT is very clear. In my opinion, there is clear different emission behaviour for both compounds in solution, which is related to the structure of the terphenyl rings. The same happens with the nido, since the nido-DT is relatively similar to closo-DT, whereas important differences are observed between closo-PT and nido-PT. In fact, no important differences are observed for closo-DT after deboronation or addition of TBAF to form nido-DT, with blue emission colour.
  •  
  • Do the authors mention that the CT-based emission can be quenched by the anionic character of the nido-o-carborane? Can you give a better explanation, i.e using TD-DFT calculations?
  • For closo-DT a low-energy emission band appears at around 482 nm in THF at 77 K. However, this band is much less intense practically priceless in the film. What is the justification for that?

Although the number of references in the introduction is quite high, my opinion is that there some missing references that should be included between references 29-64 (see list below). List of references to be included:

N. Chem. Eur. J. 2012, 18, 544

R.N. Dalton Trans. 2016, 46, 2091

Y. Chem. Commun. 2016, 52, 12494

Y. Chem. Asian J. 2017, 12, 2207

K. Org. Biomolecular chem. 2017, 15, 6913

The manuscript will be acceptable for publication in Molecules after major revision.

Author Response

Author's Reply to the Review Report (Reviewer 1).

The paper by K.M. Lee and co-authors (Reference: molecules-799158) describes the synthesis, characterization, photophysical properties and theoretical studies of four carboranyl-containing terphenyl-based compounds. They use two different terphenyl rings, one perfectly distorted to get compound closo-DT, and a planar o-terphenyl ring to get closo-PT. They analyse their emission properties in solution (solvatochromic effect and different temperatures) and film state, as well as the changes produced in the emission colour after deboronation of the closo cluster to lead to the nido-species. Similar studies have been previously addressed by the group for other carborane-based compounds. The authors have designed these compounds following a previous paper published by the group (Chem. Commun, 2019, ref 63). The synthetic procedure and the characterization of the compounds are properly done. The TD-DFT calculations match well with the experimental data.

Some minor mistakes:

-Page 5, line 226, the nido-DT is not an adduct but a salt of tetrabuthylammonium, since the nido clusters has a negative charge. Consequently, the term adduct should be removed over the manuscript.

Response: We thank the reviewer for bringing these to our attention. As requested, we have corrected “adduct” to “salt” in the revised manuscript.

- The term “decoupled” is not necessary, in 1H{11B decoupled} or 11B{1H decoupled}, you might just use 1H{11B} and 11B{1H}.

Response: As requested, we have removed the term “decoupled” from the relevant sections of the revised manuscript and supplementary material. We thank the reviewer for providing this valuable comment.

- Why NMR spectra of closo compounds are carried out in THF-d8, and nido in CD2Cl2? Is there any problem of solubility? In fact, the 11B NMR spectra of the nido compouds seem very diluted. On the other hand, the photophysical measurements are performed in THF at 298K. I wonder if authors have tried to get the NMR spectra for nido derivatives in THF-d8.

Response: We thank the reviewer for these important questions. As the reviewer suspected, both nido-o-carboranyl compounds do not dissolve well in organic solvents, presumably because they are salt-type compounds. We found that the nido-compounds were most soluble in DCM and therefore we decided to perform the NMR studies of these compounds in CD2Cl2.

Furthermore, the 1H{11B} NMR study is much better carried out in CD2Cl2 because the peaks corresponding to the B-H units of the carborane cages are more-easily quantitatively analyzed since these peaks are observed in the 2.5–1.2 ppm range in CD2Cl2. The residual protonated solvent peaks (ca. 3.6 and 1.7 ppm) overlap with those of the cages in THF-d8.

-Some doubts and question on the photophysical properties:

Why the excitation spectrum is so different to the absorption one for closo-DT?, The excitation maximum is 292 nm, whereas there are two maxima in the UV-vis spectrum (220 and 268 nm). To which transitions can be attributed the high energy band at around 220 nm that in fact is the maximum of absorption in the UV-vis of closo-DT? In the case of closo-PT, both absorption and excitation spectra match well than for closo-DT, what is the explanation?

Response: We thank the reviewer for asking these questions. In fact, the excitation spectra in Figure S13 (in the supplementary material) were acquired in order to maximize each ICT-based emission (λem = ca. 490 nm for closo-DT and 549 nm for closo-PT); i.e., to investigate the ICT-based emissive band, the excitation spectra of both closo-compounds were equally acquired. For these reasons, we believe that the absorption pattern for closo-DT centered on the LE transition (ππ*transition on the terphenyl group) is different to that of the excitation spectrum, whereas that for closo-PT, which has significant ICT-transition characteristics, is very similar to the excitation spectrum.

In fact, we can mention that the high-energy absorption bands above 220 nm do not accurately reflect these compounds, since the solvents themselves also absorb energy in this region. We believe that these absorption bands are attributable to typical ππ* transitions of the terphenyl groups.

In the table 1 there are not the fluorescence quantum yield values. Do you have determined them? In general, it is well kown that closo-o-carborane causes the quenching of the fluorescence in solution, for that reason it should be interesting to have these values for the four compounds and compare them.

Response: We very much appreciate the reviewer’s valuable comments; however, the relationship between quantum yield via the radiative pathway for the ICT transitions of terphenyl-based closo-o-carboranyl compounds and their structural features have already been reported by our group [Ref 69: Chem. Commun. 2019, 55, 14518]. For this reason, we wished to only focus on the changes in the emissive characteristics of the o-carboranyl compounds through deboronation, since the quantum yields of the nido-compounds in this manuscript are much lower than those of the closo-compounds.

Although the authors affirm that both closo derivatives (closo-DT and closo-PT) exhibit dual-emission patterns, it is very difficult for this referee (and I guess for any reader) to observe the low-energy emissions attributable to the ICT transitions for compound closo-DT in Figure 3, whereas for closo-PT is very clear. In my opinion, there is clear different emission behaviour for both compounds in solution, which is related to the structure of the terphenyl rings. The same happens with the nido, since the nido-DT is relatively similar to closo-DT, whereas important differences are observed between closo-PT and nido-PT. In fact, no important differences are observed for closo-DT after deboronation or addition of TBAF to form nido-DT, with blue emission colour.

Response: We thank the reviewer for this comment. As already mentioned in the response to the previous question, the experimental results and the relationship between various ICT-based emission behavior for the closo-compounds in the solution state, as well as the structural forms of the terphenyl groups, have been considered by us previously [Ref 69: Chem. Commun. 2019, 55, 14518]. We found out that the planarity of the terphenyl moiety gives rise to an efficient ICT-based radiative decay mechanism; hence, closo-PT shows an intense ICT-based emission at around 500 nm in the solution state at 298 K.

The new insight provided by this study is that deboronation quenches the only ICT-based emission, resulting in emissive color-change behavior in solution at 298 K for only closo-PT.

We agree with the comment made by the reviewer, that the PL spectra of closo-DT don’t appear to show dual-emissive pattern despite a weak emissive trace at around 500 nm. For that reason, we removed the term “dual” when discussing the PL spectra of closo-DT in the revised manuscript.

Do the authors mention that the CT-based emission can be quenched by the anionic character of the nido-o-carborane? Can you give a better explanation, i.e using TD-DFT calculations?

Response: In response to the reviewer’s question and suggestion, we actually performed DFT calculations on the nido-o-carboranyl compounds to obtain theoretically meaningful results. However, the data obtained did not match well with the experimental results. We presume that DFT calculations provide results for ionic compounds that are sometimes incorrect. Nonetheless, the one thing that is experimentally clear is that the ICT emissive band (which was verified by TD-DFT calculations in this study) is quenched by the deboronation of the o-carborane cage (which was verified in this study by 1H NMR spectroscopy with TBAF). For these results, we should mention that the ICT-based emission can be quenched by the anionic character of the nido-o-carborane. Such features are significantly similar to those previously reported [Ref 70: Chem. Eur. J. 2020, 26, 548].

For closo-DT a low-energy emission band appears at around 482 nm in THF at 77 K. However, this band is much less intense practically priceless in the film. What is the justification for that?

Response: Although the low-energy emission band (ICT-based emission) for closo-DT in THF at 77 K looks much more intense than that in the film state, the spectral intensities cannot be directly compared since the concentrations in both states are not the same. The most important point is that the LE-based emission (based on the ππ* transition of the terphenyl group) is significantly stronger than the ICT-based emission in any state for closo-DT; further details are discussed in our previous report [Ref 69: Chem. Commun. 2019, 55, 14518]. We wanted to verify the effect of the deboronation reaction on the photophysical properties of terphenyl-based closo-o-carboranyl compounds in this current study.

Although the number of references in the introduction is quite high, my opinion is that there some missing references that should be included between references 29-64 (see list below). List of references to be included:

Chem. Eur. J. 2012, 18, 544

R.N. Dalton Trans. 2016, 46, 2091

Chem. Commun. 2016, 52, 12494

Chem. Asian J. 2017, 12, 2207

Org. Biomolecular chem. 2017, 15, 6913

The manuscript will be acceptable for publication in Molecules after major revision.

Response: We appreciate the reviewer’s valuable suggestions. All of the suggested citations have been added to the reference section (Refs. 54-58) of the revised manuscript.

Reviewer 2 Report

This manuscript by Duk Keun An and Kang Mun Lee presents two types of para-terphenyl cores, which are flanked by two units of ortho-carboranes (closo-DT and closo-PT). The synthesis of these compounds by aryl-alkynyl coupling and boron cage formation via decaborane B10H14 is presented as well as the deboronation in the cage position B3 in the presence of fluoride. All compounds are throughly and neatly characterized for my highest pleasure.

The compounds are interesting from the standpoint of fluorescence, in particular for the closo-PT type compounds with high Stokes shift, which can be quenched upon deboronation. Computational calculations provided helpful insight and understanding of the photophysics of these systems.

I can recommend this work for publication in molecules as it stands.

Author Response

Author's Reply to the Review Report (Reviewer 2).

This manuscript by Duk Keun An and Kang Mun Lee presents two types of para-terphenyl cores, which are flanked by two units of ortho-carboranes (closo-DT and closo-PT). The synthesis of these compounds by aryl-alkynyl coupling and boron cage formation via decaborane B10H14 is presented as well as the deboronation in the cage position B3 in the presence of fluoride. All compounds are throughly and neatly characterized for my highest pleasure.

The compounds are interesting from the standpoint of fluorescence, in particular for the closo-PT type compounds with high Stokes shift, which can be quenched upon deboronation. Computational calculations provided helpful insight and understanding of the photophysics of these systems.

I can recommend this work for publication in molecules as it stands.

Response: We thank this reviewer for their favorable evaluation. It was also our pleasure.

Reviewer 3 Report

Reviewer comments to author

In the manuscript entitled “Deboronation-Induced Ratiometric Emission Variations of Terphenyl-Based Closo-o-Carboranyl Compounds: Applications to Fluoride-Sensing”, four new compounds of terphenyl-based closo-o-Carboranyl and nido-o-carboranyl compounds were prepared, characterized and luminescent properties investigated. The article (introduction section)should be shorter. However, the text is easy to read and to follow all the discussion. All the previous background of the topic is well supported by recent references. The relationship between the molecular structure of terphenyl group and their photophysical photochemical properties should be discussed in more details, and the degradation mechanism of deboronation must be explained with molecular structures. Conclusions are supported by experiments included in the text. The work seems interesting and can be published after the major revision as below:

Revisions:

  1. page 2 – line 70. Figure 1 - The molecular structures from literature are not necessary. A new Figure 1 with molecular structures of all new compounds (Nido- and Closo-o-Carboranyl Compounds) must be inserted.

  1. Page 6 – Line 241. Figure 2 – The molecular structures (closo-DT, closo-PT, nido-PT and nido-DT) must be suppressed in the chemical mechanism. Symbols and other denotations must be considered. The molecular structures were presented in Figure-1.

  1. page 2 – line 77. Delete “perfectly” (with perfectly distorted …).

  1. page 2 – line 80. Delete “perfectly” (with perfectly planar …).

  1. page 2 – line 83. Replace text: from “the o-carborane moiety could interrupt the ICT transition…” to “the o-carborane moiety may deactivate the ICT transition”.

  1. page 2 – line 84. The term color or emission change is not correct for this photophysical process. Actually, the photophysical process observed is quenching of fluorescence.

Replace text: from “and also cause the emission-color change…” to “and also quenching of the emission…”.

  1. page 6 – line 247. Replace the sentence. From “The photophysical properties of the four terphenyl-based closo- and nido-o-carboranyl 248 compounds, closo-DT, nido-DT, closo-PT, and nido-PT, were investigated…” to “ The photophysical properties of all terphenyl-based closo- and nido-o-carboranyl were investigated...”

Molecular symbols: Closo-, Nido-, DT, and -PT must be shown in new Figure 1.

  1. page 6 – line 251. The authors relates that “the bands were attributed to spin-allowed π−π* LE transitions of the central terphenylene groups and typical ICT transitions between the o-carborane 253 units and the central phenyl rings”. However, How these attributions are confirmed experimentally?

  1. Fluorescence measurements performed with THF at 77 K is observed in solid-state. Emission measurements must be performed in liquid state for different temperatures (77 and 298K).

Experimental emission measurements must be performed using methyl-ciclohexane (MCH) as solvent. Fluorescence spectra at 77K and 298 K must be compared for MCH solvent. MCH is liquid at 77 K and 298K.

  1. What is the author opinion about the following sentence?

The emission band around 500 nm is observed for all compounds. However, in frozen conditions - 77 K, the emission around 500 nm increase significantly, indicating that the electronic transition is governed by non-radiative process at 298 K.

A new paragraph in the manuscript should be written, considering the sentence above.

  1. New articles were published about photoinduced charge shifts and electron transfer in arylboron compounds. The reference section must be improved.

Author Response

Author's Reply to the Review Report (Reviewer 3).

1. page 2 – line 70. Figure 1 - The molecular structures from literature are not necessary. A new Figure 1 with molecular structures of all new compounds (Nido- and Closo-o-Carboranyl Compounds) must be inserted.

Response: We appreciate the reviewer’s valuable comment. Accordingly, the original Figure 2 has been changed to new Figure 1 in the revised manuscript.

2. Page 6 – Line 241. Figure 2 – The molecular structures (closo-DT, closo-PT, nido-PT and nido-DT) must be suppressed in the chemical mechanism. Symbols and other denotations must be considered. The molecular structures were presented in Figure-1.

Response: We appreciate the reviewer’s valuable comment. Accordingly, the synthetic routes in new Figure 1 have been simplified in the revised manuscript.

3. page 2 – line 77. Delete “perfectly” (with perfectly distorted …).

Response: We have deleted “perfectly” in the revised manuscript as suggested by the reviewer.

4. page 2 – line 80. Delete “perfectly” (with perfectly planar …).

Response: We have deleted “perfectly” in the revised manuscript as suggested by the reviewer.

5. page 2 – line 83. Replace text: from “the o-carborane moiety could interrupt the ICT transition…” to “the o-carborane moiety may deactivate the ICT transition”.

Response: We have revised this description according to the reviewer’s suggestion.

6. page 2 – line 84. The term color or emission change is not correct for this photophysical process. Actually, the photophysical process observed is quenching of fluorescence.

Replace text: from “and also cause the emission-color change…” to “and also quenching of the emission…”.

Response: We have revised this description according to the reviewer’s suggestion.

7. page 6 – line 247. Replace the sentence. From “The photophysical properties of the four terphenyl-based closo- and nido-o-carboranyl compounds, closo-DT, nido-DT, closo-PT, and nido-PT, were investigated…” to “ The photophysical properties of all terphenyl-based closo- and nido-o-carboranyl were investigated...”

Molecular symbols: Closo-, Nido-, DT, and -PT must be shown in new Figure 1.

Response: We have revised this description according to the reviewer’s suggestion. In addition, molecular designators, such as closo-, nido-, -DT, and -PT, are clearly shown in new Figure 1.

8. page 6 – line 251. The authors relates that “the bands were attributed to spin-allowed π−π* LE transitions of the central terphenylene groups and typical ICT transitions between the o-carborane units and the central phenyl rings”. However, How these attributions are confirmed experimentally?

Response: We thank the reviewer for this valuable question. In fact, p-terphenyl has the lowest absorption band at λmax = 279 nm in toluene solution; hence, this band has been assigned to a ππ* transitions of terphenylene rings [ref: J. Organomet. Chem. 2016, 825-826, 69-74]. The corresponding reference has been added to the reference section as Ref 89. Moreover, the observation in this study that both nido-compounds, neither of which can arouse an ICT transition following deboronation of the carborane cage, experimentally showed partial quenching of their lowest-energy absorption bands. Based only on the experimental results, we expect that the lowest absorption bands of all closo- and nido-compounds are assignable to ππ* transitions of terphenyl groups with substantial ICT transition involving the o-carborane.

9. Fluorescence measurements performed with THF at 77 K is observed in solid-state. Emission measurements must be performed in liquid state for different temperatures (77 and 298K).

Experimental emission measurements must be performed using methyl-ciclohexane (MCH) as solvent. Fluorescence spectra at 77K and 298 K must be compared for MCH solvent. MCH is liquid at 77 K and 298K.

Response: We appreciate the reviewer’s valuable comments. Unfortunately, all of the closo- and nido-carboranyl compounds in this manuscript are poorly soluble in non-polar solvents, especially those containing hydrocarbon chains, such as cyclohexane, n-hexane, n-pentane, petroleum ether, and so on. We checked the solubilities of these compounds in methylcyclohexane, however, as expected, they were very poorly soluble.

Furthermore, we respectfully disagree with the comment regarding the solvent states at 77 K. The melting points of THF and MCH are 164.8 K and 146.8 K, respectively. Although the melting point of MCH is a little lower than that of THF, we believe that the MCH state at 77 K is not significantly different to that of THF at 77 K. In this study, we performed PL experiments at 77 K simply in order to understand the emissive features of the rigid forms of these compounds.

For these reasons, we believe that the solvents (toluene, THF, and DCM) used in this study and the further PL experiments performed in THF at 77 K are optimal for investigating the photophysical properties of these compounds.

10. What is the author opinion about the following sentence?

The emission band around 500 nm is observed for all compounds. However, in frozen conditions - 77 K, the emission around 500 nm increase significantly, indicating that the electronic transition is governed by non-radiative process at 298 K.

A new paragraph in the manuscript should be written, considering the sentence above.

Response: We highly appreciate the reviewer’s valuable question. The suggested paragraph has been added to page 7, lines 275‒278 of the revised manuscript after minor revision.

11. New articles were published about photoinduced charge shifts and electron transfer in arylboron compounds. The reference section must be improved.

Response: We appreciate the reviewer’s suggestion. We have added the new article about photoinduced charge shifts and electron transfer in arylboron compounds ( J. Am. Chem. Soc. 2017, 139, 7681-7684) as Ref 59 in the revised manuscript.

Round 2

Reviewer 1 Report

I am satisfied with the revised paper by K.M. Lee and co-authors (Reference: molecules-799158) but, it should be nice to highlight throughout the manuscript, as well as the abstract and conclusions, that only in those fluorophores with planar terphenyl groups, as it is the case of closo-PT, the deboronation of the clusters leads to a change in the emission properties and a deactivation of the ICT-process. Then this behaviour not only depends on the anionic character of the nido cluster but also on the structure of the fluorophore.

The paper can be accepted for publication in Molecules after adding the above requirement.

Author Response

Author's Reply to the Review Report (Reviewer 1).

I am satisfied with the revised paper by K.M. Lee and co-authors (Reference: molecules-799158) but, it should be nice to highlight throughout the manuscript, as well as the abstract and conclusions, that only in those fluorophores with planar terphenyl groups, as it is the case of closo-PT, the deboronation of the clusters leads to a change in the emission properties and a deactivation of the ICT-process. Then this behaviour not only depends on the anionic character of the nido cluster but also on the structure of the fluorophore.

Response: We thank the reviewer for these important suggestions. As suggested, we tried to highlight the fact that emission properties and a deactivation of the ICT-process could be affected by the deboronation reaction of the o-carborane cages as well as the structural feature of the appended terphenyl rings. We revised some parts (highlighted in yellow) in Abstract, Introduction, description of Photophysical properties and Conclusion of the revised manuscript.

The paper can be accepted for publication in Molecules after adding the above requirement.

Response: We thank this reviewer for their favorable evaluation.

Reviewer 3 Report

The Solvent Methyl ciclohexane has melting point at 147.15 K. however, liquid phase is observed if the temperature decrease slowly. 

Indeed, the low solubility of the compounds in MCH Solvent makes Impossible to follow the photophysical properties at 77K.

my recommendation for this manuscript is accept.

Author Response

The Solvent Methyl ciclohexane has melting point at 147.15 K. however, liquid phase is observed if the temperature decrease slowly.

Indeed, the low solubility of the compounds in MCH Solvent makes Impossible to follow the photophysical properties at 77K.

my recommendation for this manuscript is accept.

Response: We appreciate the reviewer’s valuable comments. Although the compounds in this study cannot follow the photophysical properties in MCH at 77 K due to their poor solubility, we will try the experiments in further study for photophysical properties of o-carboranyl luminophores.